# Metformin Pharmacogenetics: Effects of SLC22A1, SLC22A2, and SLC22A3 Polymorphisms on Glycemic Control and HbA1c Levels

**DOI:** 10.3390/jpm9010017

**Published:** 2019-03-25

**Authors:** Laith N. AL-Eitan, Basima A. Almomani, Ahmad M. Nassar, Barakat Z. Elsaqa, Nesreen A. Saadeh

**Affiliations:** 1Department of Applied Biological Sciences, Jordan University of Science and Technology, Irbid 22110, Jordan; 2Department of Biotechnology and Genetic Engineering, Jordan University of Science and Technology, Irbid 22110, Jordan; 3Department of Clinical Pharmacy, Jordan University of Science and Technology, Irbid 22110, Jordan; baalmomani1@just.edu.jo; 4Department of Internal Medicine, MedStar Washington Hospital Center, Georgetown University, Washington, DC 20010, USA; ahmadnassar.md@gmail.com; 5Faculty of Medicine, Jordan University of Science and Technology, Irbid 22110, Jordan; barakat.saqa@gmail.com; 6Department of Internal Medicine, Jordan University of Science and Technology, Irbid 22110, Jordan; nasaadeh@just.edu.jo

**Keywords:** metformin, pharmacogenetics, solute carrier gene, SNPs, HbA1c, glycemic control

## Abstract

Type 2 diabetes mellitus (T2DM) constitutes a major portion of Jordan’s disease burden, and incidence rates are rising at a rapid rate. Due to variability in the drug’s response between ethnic groups, it is imperative that the pharmacogenetics of metformin be investigated in the Jordanian population. The objective of this study was to investigate the relationship between twenty-one single nucleotide polymorphisms (SNPs) in the *SLC22A1*, *SLC22A2*, and *SLC22A3* genes and their effects on metformin pharmacogenetics in Jordanian patients diagnosed with type 2 diabetes mellitus. Blood samples were collected from 212 Jordanian diabetics who fulfilled the inclusion criteria, which were then used in SNP genotyping and determination of HbA1c levels. The rs12194182 SNP in the *SLC22A3* gene was found to have a significant association (*p* < 0.05) with lower mean HbA1c levels, and this association more pronounced in patients with the CC genotype (i.e., *p*-value was significant before correcting for multiple testing). Moreover, the multinomial logistic regression analysis showed that SNP genotypes within the *SLC22A1*, *SLC22A2*, and *SLC22A3* genes, body mass index (BMI) and age of diagnosis were significantly associated with glycemic control (*p* < 0.05). The results of this study can be used to predict response to metformin and other classes of T2DM drugs, making treatment more individualized and resulting in better clinical outcomes.

## 1. Introduction

Diabetes mellitus (DM) refers to a heterogenous group of chronic metabolic disorders that affects the body’s ability to regulate blood glucose levels [1]. Although better understanding of the disease is creating newer classifications, DM can be broadly subdivided into type 1 DM (T1DM), where autoimmune processes cause absolute insulin deficiency, and type 2 DM (T2DM), in which a mixture of genetic and environmental factors leads to impaired insulin production [2]. By far, T2DM is the most common manifestation of the disease, accounting for up to 90% of global diabetic cases [3]. In the Arab world, the number of T2DM cases is predicted to undergo a 96.2% increase by 2035, and Jordan, with a T2DM prevalence of 17.4% as of 2008, is no exception [4,5]. Risk factors for T2DM are particularly rampant among the Jordanian population as a result of a high prevalence of metabolic syndrome, physical inactivity, obesity, cigarette smoking, and poor dietary habits [6,7,8,9,10].

The first line of treatment for T2DM is metformin, a medication that is favored for its relative lack of side effects and its excellent patient tolerance [11]. However, due to differences in individual genetic profiles, metformin does not perform equally nor optimally in all patients, leading to a reduction in the drug’s efficacy and safety [12]. Further compounding this issue is the fact that, in a study including 237 Jordanians with T2DM, more than half were observed to have poor levels of glycemic control despite metformin being a part of the majority of treatment plans [13]. As a result, discerning the genetic component underlying the variation in metformin response is necessary, especially in populations with a high prevalence of T2DM [14]. In Jordan, different clinical characteristics of diabetes have been reported between the genetically distinct Arab, Chechen, and Circassian communities, warranting different DM management and treatment protocols for each [15].

Metformin functions by reducing hepatic glucose production while simultaneously increasing peripheral glucose uptake [16]. Metformin is unique in that it does not need to undergo metabolic breakdown to affect control of blood glucose levels [17]. In order for it to decrease hepatic glucose production, however, metformin requires membrane transport proteins encoded for by solute carrier (*SLC*) genes in order to enter the cells [18]. The solute carrier family 22 member 1 (*SLC22A1*) and 3 (*SLC22A3*) genes encode the OCT1 and OCT3 proteins, respectively, which are largely responsible for hepatic and intestinal metformin uptake [19]. In addition, OCT2 (SLC22A2) is the main facilitator of metformin uptake by renal epithelial cells [20]. Various single nucleotide polymorphisms (SNPs) in the *SLC22A1*, *SLC22A2*, and *SLC22A3* genes have been found to influence metformin pharmacodynamics and pharmacokinetics, which, in turn, affect patient response to the drug [21]. 

Despite comprising a substantial proportion of Jordan’s disease burden, T2DM has been the subject of virtually no studies with regard to its genetic component and the effect of the latter on metformin response. Therefore, the aim of the present study is to address this gap in the literature by investigating the association between certain SLC22A1, SLC22A2, and SLC22A3 SNPs and metformin effectiveness, as determined by levels of glycemic control and glycohemoglobin (HbA1c), in Jordanian T2DM patients.

## 2. Materials and Methods

### 2.1. Patient Recruitment

A total of 300 subjects were approached at the Diabetes Clinic of King Abdullah University Hospital (KAUH), a tertiary referral hospital located in the north of Jordan and the primary teaching hospital for Jordan University of Science and Technology (JUST). All enrolled subjects were Jordanians previously diagnosed with diabetes who were at the Diabetes Clinic for follow-up and assessment. Subjects included in the study had an established diagnosis of T2DM according to the diagnostic guidelines set by the World Health Organization (WHO) and the American Diabetes Association (ADA). Additional inclusion criteria comprised the patients having been diabetic for at least six months prior to data collection, having regular and compliant intake of metformin as the primary pharmacologic agent in their treatment plan, and being over the age of 18. A total of 88 subjects who were not taking metformin and subjects with a failed genetic analysis (Table 1) were excluded.

### 2.2. Study Design

A comparison between T2DM patients taking metformin and different SLC22A1, SLC22A2, and SLC22A3 SNP genotypes was conducted to evaluate the effect of these genotypes on certain clinical outcomes (glycemic control and Hba1c levels). The selection process for the different SNPs was based on previous reports of their clinical and functional relevance in the field of metformin pharmacogenetics. Ethical approval to conduct the study was granted by the Ethics Committee of JUST. All procedures employed in this study were in accordance with the Declaration of Helsinki (1964). Written informed consent was obtained from all enrolled subjects after being given details of the study’s purpose and assurance of patient confidentiality.

### 2.3. Data Collection

Data collection took place between 2014 and 2015 at the Diabetes Clinic at KAUH by means of direct interviews. Clinical data was collected using a questionnaire designed for the study and filled out by healthcare personnel interviewing the subjects. Obtained clinical data included demographic details, age at diagnosis of DM, duration of DM, comorbidities, and medication history (data not shown). Blood samples for determination of HbA1c levels and DNA extraction were obtained during the interview by a phlebotomist using ethylenediamine tetraacetic acid (EDTA) collection tubes. HbA1c serum levels were measured at KAUH using standard laboratory techniques.

### 2.4. Definition of Variables

As per the ADA’s 2016 guidelines, adequate glycemic control was defined as an Hba1c level of less than 7.0%, while poor glycemic control was defined as an Hba1c level of greater than or equal to 7.0%.

### 2.5. SNP Genotyping

Genomic DNA was extracted from whole blood using a Promega kit (Promega Corporation, Madison, WI, USA). All individuals were genotyped for the twenty-one chosen SNPs listed in Table 1. Genotyping took place in Australia using the Sequenom MassARRAY system (iPLEX GOLD) (Sequenom, San Diego, CA, USA), and the manufacturer’s instructions were meticulously followed. Briefly, PCR was performed to amplify target regions, followed by treatment with shrimp alkaline phosphatase for dephosphorylation of unicorporated dNTPs. Then, the iPLEX extension reaction, which involves a single nucleotide extension of SNP sites, was carried out. Finally, matrix-assisted laser desorption/ionization-time of flight (MALDI-TOF) mass spectrometric analysis was carried out in order to detect differences in mass. The primers for the PCR and iPLEX extension reaction were designed using the Assay Design Suite. MassARRAY Workstation (v. 3.3) (Sequenom, San Diego, CA, USA) was used to process the iPLEX SpectroCHIP Bioarray (Sequenom), and Typer Analyzed (v. 4.0.2) (Sequenom) was used to analyze the genotypes obtained from the assays.

### 2.6. Statistical Analysis

Testing for consistency with the Hardy–Weinberg equilibrium was carried out using PowerMarker software (v. 3.25). Statistical analysis was performed using the Statistical Package for Social Sciences (SPSS) software (v. 22). Continuous data was presented as means ± standard deviation (SD) and categorical data was presented in numerical form. Variables were compared using Pearson’s chi-squared test, *t*-test, and ANOVA F-test as appropriate. Multinomial logistic regression analysis was applied to further test the association of SNP with glycemic control after adjusting for age, BMI and age of diabetes diagnosis as covariates. A *p*-value cutoff of <0.05 was used as a determinant of statistical significance.

## 3. Results

### 3.1. Patient Characteristics

A total of 300 diabetics were initially approached, and 212 patients were enrolled in the study after fulfilling the inclusion criteria. As can be seen from Table 2, the mean age of the studied sample was 56.64 ± 9.4 years. The mean Hba1c level was 7.93 ± 2.1%, and more than half (59.9%) of the studied subjects had inadequate glycemic control. No statistically significant difference was found between male and female subjects.

### 3.2. Minor Allelic Frequency of the Investigated OCT Candidate Gene SNPs and Their Associations with DM Treatment Response

Three *OCT* genes essential to drug metabolism were included in this study. Table 3 shows the investigated candidate SNPs within these genes and the allelic distribution frequency for each gene’s minor allele as well as the Hardy Weinberg equilibrium (HWE) *p*-value. All estimated SNP genotypes were in HWE and normally distributed with *p*-value > 0.05 except two SNPs (rs622342 and rs2928035) within *SLC22A1* gene. The present study did not find a significant association of glycemic control in T2DM patients taking metformin with seven SNPs (rs1867351, rs2282143, rs2282143, rs461473, rs4646272, rs622342, rs683369) in the *SLC22A1* gene and 10 SNPs (rs10755577, rs17588242, rs17589858, rs2928035, rs3127573, rs316024, rs316025, rs316026, rs533452, rs662301) in the *SLC22A2* gene (Table 4 and Table 5). Additionally, it was found that the rs12194182 SNP in the *SLC22A3* gene is associated with lower mean HbA1c levels in T2DM patients (*p*-value < 0.05, i.e., *p*-value was significant before correcting for multiple testing). Conversely, this SNP showed no association with adequacy or inadequacy of glycemic control as presented in Table 6. The other studied SNPs (rs2292334, rs2504927, and rs3123634) within *SLC22A3* gene exhibited no significant association with glycemic control in T2DM patients taking metformin.

### 3.3. Correlation of Glycemic Control with Age, BMI, and Age at Diabetes Diagnosis Using Multinomial Logistic Regression

The association of SNP genotype frequencies of the studied polymorphisms with glycemic control was also assessed after adjusting for the following covariates: age, BMI, and age at diabetes diagnosis (Table 7). The multinomial logistic regression analysis showed that SNP genotypes, BMI and age of diagnosis were significantly associated with glycemic control (*p*-value < 0.05). Finally, there was no significant correlation between the studied SNPs and glycemic control after adjustment for age (*p* > 0.05).

## 4. Discussion

Recent successes in identifying common variants associated with T2DM elucidated their relationship with the pathophysiology of the disease, which further aids in the evaluation of individual risk and treatment success [22]. Despite an increasingly widespread prevalence in Jordan, T2DM has not been the subject of adequate pharmacogenetic investigation in the Jordanian population. Subsequently, the present study is highly relevant in that it sheds some light on the link between variation in metformin metabolism and Jordanian genetic profiles. This study served to analyze twenty-one confirmed T2DM-predisposing variants in the *SLC22A1*, *SLC22A2*, and *SLC22A3* genes and the extent of their association with adequate glycemic control. The aforementioned genes are especially pertinent to the field of drug transport because they encode the OCT proteins, which are organic cation transporters that play key roles in the regulation of essential metabolic pathways [23,24].

OCT1, encoded by the *SLC22A1* gene, is responsible for the bulk of hepatic metformin uptake [25]. Pharmacogenetic studies on mice revealed that mice with a knockout *OCT1* gene exhibited lower hepatic concentrations of metformin in addition to an impaired glucose-lowering effect [26]. Healthy subjects with reduced OCT1 function due to R61C, G4015, 420 del, or G46512 polymorphisms have shown a profound effect on metformin pharmacokinetics, indicating that the OCT1 genotype is a determinant of the latter [27,28]. Another study showed that the rs187351, rs4709400, rs628031, and rs2297374 SNPs affect glycemic outcomes after metformin treatment in Han Chinese T2DM patients [29]. However, a study conducted in the Caucasian population found that only the rs622342 SNP was associated with glycemic outcome [30]. On the contrary, a recent meta-analysis concluded that none of the *SLCA22A1* SNPs had any significant effect on glycemic response or HbA1c levels in T2DM patients [31]. The results of the current study show no significant association between glycemic outcomes after metformin treatment and the rs622342 SNP or any of the other studied SLC22A1 SNPs shown in Table 4. This observation indicates that these polymorphisms have no effect on HbA1c levels in Jordanian T2DM patients taking metformin. 

The *SLC22A2* gene encodes for the OCT2 protein, which facilitates the transport of metformin from the bloodstream into the renal epithelial cells [30]. Genetic variants in the *SCL22A2* gene, such as T199I, T201M, and A270S, have been associated with an increased plasma concentration and a decreased renal clearance of metformin [26]. Recent studies have also found that compounds with a guanidine group like metformin are better substrates for OCT2 in mice and humans [32]. In fact, the OCT2 gene variant 808 G>T showed a profound effect on metformin pharmacokinetics in healthy subjects by exhibiting an association with higher plasma concentrations [32]. Additionally, the 808 G<T polymorphism also demonstrated reduced renal metformin clearance in healthy Chinese subjects [33]. However, no significant association between renal metformin clearance and certain SLC22A2 SNPs (rs10755577, rs17588242, rs17589858, rs2928035, rs312024, rs312025, rs312026, rs3127573, rs533452, and rs662301) was detected in healthy Caucasian males [34]. However, the current study did not find any statistical significance to show that any of the studied SLC22A2 SNPs shown in Table 5 to have any effect on glycemic control.

The OCT3 protein, which is coded for by the *SLC22A3* gene, is thought to be involved in metformin uptake into hepatocytes and the interstitial fluid [19]. In healthy male Caucasians, no statistically significant association between four SLC22A3 SNPs (rs12194182, rs2292334, rs2504927, and rs3123634) and metformin pharmacokinetics was found [34]. It has also been reported that the rs2292334 SNP is associated with a decreased risk of T2DM and a decrease in HbA1c levels [35]. The current study concluded that the rs12194182 SNP in the *SLC22A3* gene is linked to lower mean HbA1c levels in the Jordanian T2DM patients. Subjects with the CC genotype exhibited the lowest mean HbA1c levels, while patients with the CT and TT genotypes exhibited higher levels. However, the other studied SNPs (rs2292334, rs2504927, and rs3123634) were in accordance with the findings of Tzvetkov et al. [34] as no significant link was established between these SNPs and glycemic control or mean HbA1c levels. These reports, in addition to the results of the present study, show that OCT3 might be somewhat associated with metformin’s effect on HbA1c levels.

Finally, after adjusting for BMI and age at diabetes diagnosis using multinomial logistic regression, this study found a genetic association between glycemic control and all tested SNPs within *SLC22A1*, *SLC22A2* and *SLC22A3* genes (Table 2 and Table 7). The differences in BMI values between patients or the differences in effect size between different populations might be the reasons why the effect of the aforementioned SNPs could not be replicated in the current study. The variability in age at diabetes diagnosis also has a major effect on the genetic association of these SNPs with glycemic control in the treatment of diabetes. These covariate factors should be considered when treating patients with diabetes. It is also important to clarify the impact of these factors on the genetic associations with glycemic control in the T2DM population.

One potential limitation of the present study is that the duration of the diagnosis was not considered, and subjects who had the disease for a longer time could have decreased production of endogenous insulin, meaning that levels of endogenous insulin among the subjects were variable. Another limitation to be considered is that not all patients were taking the same dosage of metformin, and no baseline levels of HbA1c were recorded to study the degree in which these levels were affected by metformin monotherapy. Most importantly, the relatively small sample size could limit the ability to extrapolate results to the general population. However, it is important to note that the present study is the first to investigate the association between the aforementioned SLC22A SNPs and TD2M in the Jordanian population. 

## 5. Conclusions

The present study revealed that the rs12194182 SNP of the *SLC22A3* gene was associated with better HbA1c levels. However, this SNP was not associated with glycemic control after applying the multiple comparison analysis with *p* value > 0.002. Owing to the high prevalence of the disease and its complications among the Jordanian population, the results of this study might provide great benefits for patients upon the introduction of personalized T2DM therapy in Jordan. The initiation of early individual-based treatment would aid in disease prognosis and hopefully lead to lower rates of microvascular and macrovascular T2DM complications due to better and more focused treatment regimens.

## Figures and Tables

**Table 1 jpm-09-00017-t001:** List of studied SNPs, chromosomal positions, gene locations based on the National Center for Biotechnology Information (NCBI) Human Genome Assembly Build 36.3 and diabetes mellitus (DM) patients fail to genotype.

Gene	SNP ID	Chromosomal Position	SNP	Gene Location	DM Patients Fail to Genotype
***OCT1***	rs1867351	6:160543123	T>C	Exon 1	2
	rs2282143	6:160557643	C>T	Exon 3	0
	rs2297374	6:160551204	C>T	Intron 9	1
	rs461473	6:160543562	G>A	Intron 1	0
	rs4646272	6:160551093	T>G	Intron 1	1
	rs622342	6:160572866	A>C	Intron 9	55
	rs683369	6:160551204	C>G	Exon 2	0
***OCT2***	rs10755577	6:160219462	C>T	Intron 10	1
	rs17588242	6:160242199	C>T	Intron 8	8
	rs17589858	6:160268084	G>A	Promoter	9
	rs2928035	6:160560871	A/G	Intron 10	16
	rs3127573	6:160681393	A/G	Promoter	47
	rs316024	6:160682236	A>G	Promoter	0
	rs316025	6:160603371	C>T	Promoter	1
	rs316026	6:160604360	T/C	Promoter	1
	rs533452	6:160276730	C>T	Promoter	20
	rs662301	6:160696919	C>T	Promoter	0
***OCT3***	rs12194182	6:160834515	C/T	Intron 5	15
	rs2292334	6:160858188	G>A	Exon 7	3
	rs2504927	6:160780420	G>A	Intron 7	18
	rs3123634	6:160381207	C>T	Intron 1	3

**Table 2 jpm-09-00017-t002:** Clinical characteristics of Jordanian patients with diabetes.

Category	Subcategory	Adequate Glycemic Control	Inadequate Glycemic Control ^b^	*p*-Value ^a^
**Gender**	Male	30 (14.2%)	52 (24.5%)	0.472
	Female	55 (25.9%)	75 (35.4%)	
**Age**		57.17 ± 9.7	56.30 ± 9.3	0.504
**BMI ^c^ (kg/m^2^)**		31.18 ± 5.1	33.16 ± 6.4	0.023
**Age of diabetes diagnosis**		49.46 ± 9.2	46.74 ± 9.6	0.042
**Glycemic Parameters**	Fasting glucose	6.55 ± 1.8	9.44 ± 4.5	7.3 × 10^−8^
	HbA1c ^d^	6.27 ± 1.3	9.08 ± 1.8	5 × 10^−26^
	HDL ^e^ cholesterol (mmol/L)	1.21 ± 0.3	1.10 ± 0.3	0.010
	LDL ^f^ cholesterol (mmol/L)	2.80 ± 0.9	3.03 ± 0.8	0.024
	Cholesterol (mmol/L)	4.30 ± 1.0	4.50 ± 1.1	0.178
	Triglycerides (mmol/L)	1.69 ± 1.2	2.10 ± 1.7	0.530
	Serum creatinine (µmol/L)	74.14 ± 24.3	75.92 ± 25.7	0.613
	Creatinine clearance (mL/min)	109.82 ± 35.0	118.80 ± 42.0	0.105

^a^*p*-value < 0.05 is considered significant, ^b^ defined as HbA1c level ≥7.0% according to the American diabetic association (ADA) guidelines, ^c^ BMI: Body mass index, ^d^ HbA1c: glycated haemoglobin, ^e^ HDL: High-density lipoprotein, ^f^ LDL: Low-density lipoprotein.

**Table 3 jpm-09-00017-t003:** Minor allele frequencies among DM patients and the HWE *p*-value of candidate polymorphisms in *OCT1*, *OCT2* and *OCT3* genes.

Gene	SNP ID	MA ^a^	MAF ^b^	χ2	HWE ^c^ *p*-Value
***OCT1***	rs1867351	G	0.19	0.04	0.85
	rs2282143	T	0.02	0.06	0.81
	rs2297374	C	0.46	0.07	0.79
	rs461473	A	0.10	0.60	0.44
	rs4646272	G	0.04	1.40	0.24
	rs622342	C	0.23	4.03	0.04
	rs683369	G	0.13	0.79	0.37
***OCT2***	rs10755577	T	0.18	0.29	0.59
	rs17588242	C	0.25	0.01	0.92
	rs17589858	G	0.25	0.01	0.94
	rs2928035	G	0.19	4.96	0.03
	rs3127573	C	0.08	0.04	0.85
	rs316024	A	0.21	3.69	0.06
	rs316025	A	0.24	2.50	0.11
	rs316026	T	0.42	2.38	0.12
	rs533452	T	0.29	0.05	0.82
	rs662301	T	0.05	0.25	0.61
***OCT3***	rs12194182	C	0.09	1.12	0.29
	rs2292334	T	0.28	1.02	0.32
	rs2504927	G	0.48	1.26	0.26
	rs3123634	T	0.37	2.28	0.13

^a^ MA: minor allele. ^b^ MAF: minor allele frequency. ^c^ HWE: Hardy–Weinberg equilibrium.

**Table 4 jpm-09-00017-t004:** Effect of genotype distribution of studied SNPs for *OCT1* gene on glycemic control and on HbA1c level in Jordanian T2DM patients receiving metformin.

Gene	SNP ID	Genotype	Total	Adequate Glycemic Control	Inadequate Glycemic Control	*p*-Value *	Mean HbA1c ± SD	*p*-Value *
***OCT1***	rs1867351	AA	138	58	80	0.187	7.93 ± 2.06	0.136
		GA	65	21	44		8.21 ± 2.04	
		GG	7	5	2		6.45 ± 2.33	
	rs2282143	CC	205	84	121	0.153	7.98 ± 2.11	0.872
		CT	7	1	6		8.11 ± 1.48	
		TT	0	0	0		-	
	rs2297374	CC	46	21	25	0.285	7.76 ± 1.80	0.180
		TC	103	43	60		7.83 ± 2.02	
		TT	62	20	42		8.43 ± 2.36	
	rs461473	AA	1	1	0	0.311	5.60	0.253
		GA	39	13	26		8.36 ± 2.07	
		GG	172	71	101		7.91 ± 2.09	
	rs4646272	GG	1	0	1	0.074	10.90	0.277
		GT	15	2	13		8.48 ± 2.28	
		TT	195	82	113		7.94 ± 2.07	
	rs622342	AA	88	33	55	0.432	8.14 ± 2.20	0.277
		CA	65	24	41		8.16 ± 2.17	
		CC	4	1	3		8.10 ± 2.09	
	rs683369	CC	160	66	94	0.146	7.83 ± 1.89	0.072
		GC	50	17	33		8.52 ± 2.61	
		GG	2	2	0		6.45 ± 0.48	

* *p*-value < 0.05 is considered significant.

**Table 5 jpm-09-00017-t005:** Effect of genotype distribution of studied SNPs for *OCT2* gene on glycemic control and on HbA1c level in Jordanian T2DM patients receiving metformin.

Gene	SNP ID	Genotype	Total	Adequate Glycemic Control	Inadequate Glycemic Control	*p*-Value *	Mean HbA1c ± SD	*p*-Value *
***OCT2***	rs10755577	TT	8	5	3	0.222	7.16 ± 1.13	0.490
		TC	60	40	20		8.20 ± 2.17	
		CC	143	84	59		7.94 ± 2.10	
	rs17588242	CC	13	7	6	0.105	7.91 ± 2.16	0.221
		CT	78	32	46		8.14 ± 2.21	
		TT	113	40	73		7.98 ± 2.09	
	rs17589858	CC	114	48	66	0.772	7.94 ± 2.04	0.628
		CG	76	27	49		8.08 ± 2.13	
		GG	13	6	7		8.31 ± 2.14	
	rs2928035	AA	133	47	86	0.152	8.08 ± 2.09	0.237
		AG	51	22	29		8.07 ±2.03	
		GG	12	6	60		7.98 ± 2.09	
	rs3127573	CC	1	1	0	0.552	6.30	0.502
		CT	26	12	14		7.60 ± 1.72	
		TT	138	71	112		8.03 ± 2.13	
	rs316024	AA	21	10	11	0.455	7.93 ± 2.50	0.349
		AG	73	32	41		7.71 ± 1.94	
		GG	118	43	75		7.98 ± 2.09	
	rs316025	AA	16	7	9	0.641	8.12 ± 2.52	0.798
		GA	68	26	42		7.84 ± 1.93	
		GG	127	51	76		7.98 ± 2.09	
	rs316026	CC	76	28	48	0.694	8.16 ± 2.07	0.559
		CT	92	40	52		7.77 ± 2.07	
		TT	43	17	26		7.98 ± 2.09	
	rs533452	CC	97	38	59	0.797	8.17 ±2.10	0.308
		TC	78	31	47		7.84 ±2.05	
		TT	17	6	11		8.25 ±2.33	
	rs662301	CC	190	77	113	0.385	7.97 ± 2.09	0.677
		CT	21	7	14		8.15 ± 2.16	
		TT	1	1	0		6.30	

* *p*-value < 0.05 is considered significant.

**Table 6 jpm-09-00017-t006:** Effect of genotype distribution of studied SNPs for *OCT3* gene on glycemic control and on HbA1c level in Jordanian T2DM patients receiving metformin.

Gene	SNP ID	Genotype	Total	Adequate Glycemic Control	Inadequate Glycemic Control	*p*-Value *	Mean HbA1c ± SD	*p*-Value *
***OCT3***	rs12194182	CC	3	2	1	0.167	6.90 ± 1.05	0.007
		CT	31	8	23		8.94 ± 2.33	
		TT	163	67	96		7.93 ± 2.06	
	rs2292334	CC	112	46	66	0.974	7.95 ± 2.11	0.762
		CT	78	30	48		8.04 ± 2.30	
		TT	19	8	11		8.15 ± 1.79	
	rs2504927	AA	57	22	35	0.526	7.90 ± 1.68	0.232
		GA	89	33	56		8.16 ± 2.39	
		GG	48	20	28		8.08 ± 1.93	
	rs3123634	CC	77	35	42	0.495	7.62 ± 1.91	0.230
		CT	108	38	70		8.25 ± 2.03	
		TT	24	11	13		7.87 ± 2.38	

* *p*-value < 0.05 is considered significant.

**Table 7 jpm-09-00017-t007:** Associations of SNPs with glycemic control in multinomial logistic regression after adjustment for age, BMI and age at diabetes diagnosis.

Gene	SNP ID	Age *	BMI *	Age at Diabetes Diagnosis *
***OCT1***	rs1867351	0.329	0.025	0.028
	rs2282143	0.268	0.017	0.027
	rs2297374	0.323	0.021	0.033
	rs461473	0.352	0.019	0.043
	rs4646272	0.246	0.025	0.038
	rs622342	0.092	0.002	0.036
	rs683369	0.226	0.048	0.019
***OCT2***	rs10755577	0.375	0.014	0.056
	rs17588242	0.332	0.021	0.035
	rs17589858	0.174	0.019	0.024
	rs2928035	0.176	0.003	0.036
	rs3127573	0.391	0.025	0.040
	rs316024	0.353	0.016	0.039
	rs316025	0.282	0.012	0.045
	rs316026	0.315	0.013	0.022
	rs533452	0.320	0.008	0.058
	rs662301	0.308	0.009	0.035
***OCT3***	rs12194182	0.214	0.021	0.035
	rs2292334	0.243	0.013	0.023
	rs2504927	0.284	0.010	0.050
	rs3123634	0.317	0.018	0.022

* *p*-value < 0.05 is considered significant.

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
