# Peer review of "Metformin Pharmacogenetics: Effects of SLC22A1, SLC22A2, and SLC22A3 Polymorphisms on Glycemic Control and HbA1c Levels"

_jpm, 2019, doi:10.3390/jpm9010017_

Reviewer 1 Report

The manuscript by Eitan et al. sought to address a relevant issue in diabetic field that is understanding the role of metformin.

This interest in this issue derives also from the poor knowledge on the actual molecular target of metformin and, hence, on its moelcular mechanism.

However I have a major concern on the design/development of the study which shows, as main result, the association of the SNP 12194182 of OCT3 to better glycaemic control in patients taking metformin as anti-diabetic drug.  This result that is interesting per se is not well inserted in a more complex scenario that would help readers understanding the relevance of such SNP/association to metformin.

Therefore, besides some writing issues and few concerns listed below, I would re-write and re-consider the discussion of the results including a focus on:

- the relevance of studying Jordan population and its genetic/environmental features.

- The relevance of OCTs in the field of drug transport by referring to reviews in the field (few examples are: Koepsell H et al. Mol Aspects Med. (2013); Nigam SK. Annu Rev Pharmacol Toxicol. 2018; Pochini et al. SLAS Discov. 2019 etc.);

- The link of metformin distribution to improved diabetic treatments, where possible considering the main pitfalls that authors include in the conclusions of their work. I would strongly suggest to re-write conclusions in order to remove the description of limitations of the work that, instead, should be afforded in the discussion. On the contrary, I would expect in the conclusions the descriptions of the main achievements of the work in terms of translation to clinics.

Minor points:

- in Materials and Methods authors state that they follow meticulously the instructions of manufactures. However, some more details should be given on the used protocols.

- Few english typos are present.

Author Response

Dear Reviewer,

            I would like to extend my deepest thanks to you for your constructive comments and suggestions with regard to the manuscript titled “Metformin Pharmacogenetics: Effects of SLC22A1, SLC22A2, and SLC22A3 Polymorphisms on Glycemic Control and HbA1c Levels”. I am pleased to submit the revised version of the paper that addresses each comment noted by you. The manuscript was also reviewed by an English language editor in order to enhance its flow and scientific content.

Comments by Reviewer #1

1.       Therefore, besides some writing issues and few concerns listed below, I would re-write and re-consider the discussion of the results including a focus on:

i.        the relevance of studying Jordan population and its genetic/environmental features

Added to discussion.

ii.      The relevance of OCTs in the field of drug transport by referring to reviews in the field (few examples are: Koepsell H et al. Mol Aspects Med. (2013); Nigam SK. Annu Rev Pharmacol Toxicol. 2018; Pochini et al. SLAS Discov. 2019 etc.);

Added to discussion.

iii.    The link of metformin distribution to improved diabetic treatments, where possible considering the main pitfalls that authors include in the conclusions of their work. I would strongly suggest to re-write conclusions in order to remove the description of limitations of the work that, instead, should be afforded in the discussion. On the contrary, I would expect in the conclusions the descriptions of the main achievements of the work in terms of translation to clinics.    

Removed limitations from the conclusions and added them to discussion. Conclusion already mentions clinical translation of results.

2.       in Materials and Methods authors state that they follow meticulously the instructions of manufactures. However, some more details should be given on the used protocols.

Elaborated in more detail on used protocols.

3.       Few english typos are present

Corrected

Reviewer 2 Report

The manuscript by AL-Eitan et. al examines the influence of common polymorphisms in metformin transporter genes on the response of metformin therapy in a group of type 2 diabetes patients. Although the metformin pharmacogenetics is an interesting topic the manuscript suffers from the number of weaknesses at the design and experimental level depicted in comments below.   

 Specific comments:

 The study group and its selection are not sufficiently presented in the paper. According to good practice, one should provide the number of those failed to be included after applying each exclusion criteria. For example, the failing at genetic analysis is mentioned as an exclusion criteria without mentioning the number of patients for whom the genotyping failed.

Similarly, the Table1 providing information on patient characteristics is very limited on the amount of information. According to methods information on the age at diagnosis of DM, duration of DM, comorbidities, and medication history have been collected but is not displayed in Table 1. Also, BMI and dose of metformin as well as information on co-medication frequency are not given.

Authors should describe the principles for selection of the particular SNPs including the specific details.

Knowing the relatively small sample size for this T2D cohort, the proper power calculations should be presented in order to understand the limitations of the study.

The estimated frequency of each SNP in the studied cohort is not given.

A number of issues concerning the statistical analysis:

First, it is not clear what genetic model was considered in the statistical analysis in the case of categorical variables. It would be reasonable to perform the logistic regression analysis comparing the genotype occurrence between responders and non-responders, including other measures (age, sex duration of therapy, BMI etc.) as covariates. In case of continuous variable HbA1c linear regression with the same co-factors would be more appropriate.

Knowing the number of SNPs tested the authors should have to adjust the significance for multiple testing and change the conclusions on the significance of association accordingly.

It is confusing to read in the results: “As presented in Tables 3 and 4, this study found a significant association of glycemic control in T2DM patients taking metformin with seven and ten SNPs….” even though none of the p-values in Tables 3 and 4 are below 0.05.

In Table 1. It seems that columns have been switched (“SNP”, “Gene Location” and “Chromosomal Position”).

Author Response

Dear Reviewer,

            I would like to extend my deepest thanks to you for your constructive comments and suggestions with regard to the manuscript titled “Metformin Pharmacogenetics: Effects of SLC22A1, SLC22A2, and SLC22A3 Polymorphisms on Glycemic Control and HbA1c Levels”. I am pleased to submit the revised version of the paper that addresses each comment noted by you. The manuscript was also reviewed by an English language editor in order to enhance its flow and scientific content.

Comments by Reviewer #2

1.       The study group and its selection are not sufficiently presented in the paper. According to good practice, one should provide the number of those failed to be included after applying each exclusion criteria. For example, the failing at genetic analysis is mentioned as an exclusion criteria without mentioning the number of patients for whom the genotyping failed.

Information related to the exclusion criteria was added to the section 2.1 and the number of patients fails to genotype for each SNP was added to Table 1. 

2.       Similarly, the Table1 providing information on patient characteristics is very limited on the amount of information. According to methods information on the age at diagnosis of DM, duration of DM, comorbidities, and medication history have been collected but is not displayed in Table 1. Also, BMI and dose of metformin as well as information on co-medication frequency are not given.

Added the phrase “Data not shown” in the Methods section, as this data will be employed in a future study.

3.       Authors should describe the principles for selection of the particular SNPs including the specific details.

Added description of how SNPs were selected based on previously reported clinical and functional relevance in the field of metformin pharmacogenetics.

4.       Knowing the relatively small sample size for this T2D cohort, the proper power calculations should be presented in order to understand the limitations of the study.

Calculating power retrospectively is advised against by statisticians as it often leads to , especially in cases where negative results are reported [1,2]. However, we followed Kim and Seo’s (2013) guidelines on dealing with smaller sample sizes, which were to “use continuous variables rather than nominal variables”, “reduce standard deviation by precise and exact estimation of the continuous variable”, and to “set common and distinct variables as primary outcomes” [3]. Nonetheless, the limitations of the relatively small sample size were added to the discussion.

5.       The estimated frequency of each SNP in the studied cohort is not given.

Table 3 added and shows the investigated candidate SNPs within these genes and the allelic distribution frequency for each gene’s minor allele as well as the HWE p-value.                                       

6.       First, it is not clear what genetic model was considered in the statistical analysis in the case of categorical variables. It would be reasonable to perform the logistic regression analysis comparing the genotype occurrence between responders and non-responders, including other measures (age, sex duration of therapy, BMI etc.) as covariates. In case of continuous variable HbA1c linear regression with the same co-factors would be more appropriate.

Categorical values were not directly evaluated in the present study. Association with inadequate glycemic control (categorical) was calculated via association with HbA1c (continuous).

7.       Knowing the number of SNPs tested the authors should have to adjust the significance for multiple testing and change the conclusions on the significance of association accordingly.

Corrected and the conclusion has been changed

8.       It is confusing to read in the results: “As presented in Tables 3 and 4, this study found a significant association of glycemic control in T2DM patients taking metformin with seven and ten SNPs….” even though none of the p-values in Tables 3 and 4 are below 0.05.

Rephrased to make the aforementioned result clearer.

9.       In Table 1. It seems that columns have been switched (“SNP”, “Gene Location” and “Chromosomal Position”). 

Corrected.

1.        Hoenig, J.M.; Heisey, D.M. The Abuse of Power. Am. Stat. 2001, 55, 19–24.

2.        O’Keefe, D.J. Brief Report: Post Hoc Power, Observed Power, A Priori Power, Retrospective Power, Prospective Power, Achieved Power: Sorting Out Appropriate Uses of Statistical Power Analyses. Commun. Methods Meas. 2007, 1, 291–299.

3.        Kim, J.; Seo, B.S. How to calculate sample size and why. Clin. Orthop. Surg. 2013, 5, 235–42.

Round  2

Reviewer 1 Report

Authors have dealt with all the concerns.

Author Response

We thank you for your comments and careful critique of the manuscript.

Reviewer 2 Report

The manuscript has been improved significantly by the authors and the majority of the comments are answered.

However, the fact that significant characteristics of the study group are not presented in the manuscript and not included in the statistical analysis still remain open. From my viewpoint, the argument that: “data will be employed in a future study” is not valid to avoid including these data in the current study. The age at diagnosis of DM, duration of DM, comorbidities, and medication history as well as BMI and dose of metformin may have significant effect on the association of the genetic factors analysed in this study. Therefore these data must be presented in the characteristics of the studied cohort (e.g. Table 1.) and depending on their effect on the outcome, these co-variates should also be included in statistical analysis.

Author Response

Dear Reviewer,

We thank you for your comments and careful critique of the manuscript. The current submission has been revised based on your constructive comments and suggestions. Below we detail our point-by-point response to the comments and suggestions. We hope that we have adequately addressed all concerns and that the manuscript will be accepted for publication in your respective journal.

The manuscript has been improved significantly by the authors and the majority of the comments are answered. However, the fact that significant characteristics of the study group are not presented in the manuscript and not included in the statistical analysis still remain open. From my viewpoint, the argument that: “data will be employed in a future study” is not valid to avoid including these data in the current study. The age at diagnosis of DM, duration of DM, comorbidities, and medication history as well as BMI and dose of metformin may have significant effect on the association of the genetic factors analysed in this study. Therefore these data must be presented in the characteristics of the studied cohort (e.g. Table 1.) and depending on their effect on the outcome, these co-variates should also be included in statistical analysis.

We thank you for these comments and suggestions, all suggested comments were considered, described and discussed in the revised manuscript.

These comments were considered in the following tables:

Table 2: Clinical Characteristics of Jordanian patients with diabetes

Table 7: Associations of SNPs with glycemic control in multinomial logistic regression after adjustment for age, BMI and age at diabetes diagnosis.

Round  3

Reviewer 2 Report

The manuscript has been improved significantly by the authors and the comments are answered.